# Sustainability Reporting for Inland Port Managing Bodies: A Stakeholder-Based View on Materiality

**Magali Geerts * and Michaël Dooms**

Department of Business, Vrije Universiteit Brussel, B-1050 Brussels, Belgium; michael.dooms@vub.be
* Correspondence: magali.geerts@vub.be

**Abstract:** Sustainability reporting has been marked by a rise in importance in recent years as it has proved to be an important management tool in the understanding of where an organization is situated along the sustainability pathway. However, industries have shown different behaviors towards embracing this practice. In this paper, we turn our attention to the port industry, using the metropolitan inland Port of Brussels (Belgium) as a case study. Given the contested nature of port activities within urban regions, metropolitan inland ports are expected to benefit from the development of a sustainability report as it allows a more transparent account of the contribution of port activities to the objectives of different stakeholder groups in the urban environment. The case study is based on a survey yielding 74 valid responses from different stakeholder groups (employees, clients, and broader society). Our results show that the expected content of a sustainability report is viewed differently by these various stakeholder groups in terms of the relative importance of the dimensions of the Triple Bottom Line (TBL), as well as in terms of the specific indicators representing material issues. Furthermore, the concept of boundary setting with respect to the different dimensions of the TBL and the desired level of inclusion by stakeholders during the development of a sustainability report are differently assessed. The paper is of interest to academics as well as policy makers, as the research results complement the existing insights on sustainability reporting in general and can be used as basis to stimulate the adoption of sustainability reporting by inland ports.

**Keywords:** sustainability reporting; inland ports; Triple Bottom Line; materiality analysis; stakeholder management; boundary setting

---

## 1. Introduction

In the present business environment, organizations are under pressure by diverse groups of stakeholders to pay attention to 'sustainability' when they report about their performance. Existing literature and recent research show that the term 'sustainability' and associated management and reporting practices are mostly centered around three large domains, namely economic, social, and environmental—in other terms the Triple Bottom Line or TBL dimensions (people, planet, and profit). From a historical perspective, the end of the 19th century was characterized by the emergence of the practice of financial reporting. About a century later, the importance of social and environmental aspects linked to business activities had increased and hence stimulated organizations to publish additional nonfinancial information [1]. Since then, a remarkable increase in the availability of information related to the (positive/negative) effects of an organization's operations in terms of both social and environmental aspects can be observed [2,3]. Most of the largest global companies, i.e., 93% of the top 250 companies listed in the Fortune Global 500 ranking (also called the G250) [4], are investing time and resources to report and communicate on what can be defined as sustainable performance [3], which is reflected in an increase in the practice of sustainability reporting of almost 60% since KPMG's first survey on the topic in 1999. The TBL concept is often used as a framework to operationalize the

content of a sustainability report [5], as it is based on the same three dimensions (economic, social, and environmental) that are required in a sustainability report, as stated by the Global Reporting Initiative (GRI) definition [6].

While initially multinational corporations have been central in the sustainability debate, as well as in scientific research around the topic, the scope has recently been broadened to organizations in general. However, international reporting guidelines and standards (such as the GRI), as well as codes of conduct, are still largely built upon the features of larger organizations. Due to this development, there exists a reporting gap between large multinationals and small–medium enterprises (SMEs) [7], as the latter category is constrained by human and financial resources [8]. In general, SMEs continue to struggle with finding the balance between the value that arises from sustainability initiatives, of which sustainability reporting is one, and the organizational costs linked to it [9]. Furthermore, research on sustainability reporting is largely centered around the private (for profit) sector. On the other hand, public sector organizations, fully or partially owned by the government, may face stronger accountability expectations and obligations, as they are charged with tasks covering holistic thinking on social welfare and justice of the public [10,11]. As a result, public sector organizations and state-owned enterprises (SOEs) are more and more stimulated, and even legally required in some countries (e.g., Sweden) to disclose sustainability performance information.

For the research presented in this paper, we turn our attention to the port industry, which is characterized by the presence of public sector organizations. More specifically, we focus on inland ports (as defined by Rodrigue and Notteboom [12]), and the often (partially) government owned port managing bodies (PMBs)—the use of 'port authority' diminishes because of the increase of many new governance models, covering more responsibilities than only those of an authority [13]. Those PMBs are responsible for coordinating, regulating, and developing the economic activities by, inter alia, allocating land to port users (mostly private companies) to carry out logistics and industrial activities using inland waterways, while at the same time monitoring both public and private values [14]. Inland ports operate under different, sometimes more extreme circumstances than seaports, specifically when it comes to their relatively small organizational size (in terms of human and financial resources) and their location in densely populated areas. Given the widespread positive and negative externalities associated with port activities, inland PMBs are confronted with stakeholder pressure to improve the 'sustainability' footprint of the logistics and industrial areas they manage. This creates specific challenges in the context of sustainability reporting implementation, as the growing complexity of the practice has not yet been equated with a similar degree of methodological sophistication [15]. In the specific context of inland ports, there exists a need to better understand the perspectives of different stakeholder groups, both on the general concept of sustainability (reporting) and the related methodological aspects such as materiality, boundary setting, and the need for stakeholder inclusion. Furthermore, existing research is often focused on the insights of the postimplementation phase of the practice of sustainability reporting. This paper provides an alternative approach by putting the focus on a case study that is yet to start with the development of a sustainability report. Hence, this research aims to provide knowledge on expectations and needs of inland PMBs' stakeholders, stimulating the wider industry to start reporting by partially bridging the often mentioned barrier of the lack of resources and knowledge.

The paper is organized as follows. Section 2 starts by providing an overview of the current situation of sustainability reporting in the port industry, followed by a discussion on important methodological topics related to sustainability reporting and for which no equivocal answer exists at present: TBL concept, boundary setting, materiality, and stakeholder inclusion. Section 3 contains the methodology. Section 4 provides an overview and interpretation of the survey results and Section 5 provides a synthesis of the literature insights and survey results, respectively the theoretical and practical perspective on the discussed topics. A general conclusion and limitations of the research as well as suggestions for future research are presented in Section 6.

## 2. Literature Review

### 2.1. Sustainability Reporting in the Port Industry

According to Lynch et al. [3], sectors that have the largest (negative) influence on the environment and society (e.g., oil and gas, mining, etc.) tend to show larger efforts concerning sustainability initiatives, with sustainability reporting being part of it. For example, the airport sector has developed sector-specific guidelines under the auspices of the GRI, providing a practical framework to work with. However, due to the nonexistence of sectoral guidelines for ports and their managing bodies, the port industry has not yet reached the same level as other subsectors in the transportation industry. Nevertheless, in recent years, PMBs have become aware of their role and responsibility in the global transportation supply chain with respect to their environmental and social performance. In the context of seaports, PMBs are increasingly including sustainability reporting into their long-term development strategy [16], as they are not only aware of the benefits linked to it, but also because they mostly possess the necessary financial and human resources to do so. However, in general, the industry still shows a large variety in approaches. This is due to significant differences in terms of the type and profile of port activities (e.g., industrial and logistics oriented seaports, cruise ports, inland ports, and fishing ports), which leads to different (or even no) approaches concerning sustainability reporting [17,18]. Since 2015, as a trade association initiative, some larger seaports have installed a specific working group—International Association of Ports and Harbors—Permanent International Commission for Navigation Congresses (IAPH-PIANC) WG174 Sustainability Reporting for Ports—with participation of seaport representatives, consultants, and academics. This working group aims to establish port-specific sustainability reporting standards to create a more harmonized reporting framework given the multiple approaches used by the PMBs [18].

In contrast to seaports, sustainability reporting is still largely undiscovered territory for inland ports. Furthermore, as stated by Vejvar, Lai, Lo, and Fürst [19] "even though inland port operators strive for economic viability, there are growing pressures from various stakeholders for continuous enhancement of their environmental and social sustainability practices". However, no inland port or dry port, to our knowledge, has already published a sustainability report. Inland ports, given their relative smaller size in terms of operations, are considered the smaller players within the port sector [19]. They are also subject to more stringent internal financial and human constraints, and operate in an environment of high external stakeholder pressure. At the same time, their importance increases as they are a key element for sustainable port system development, given their role in new patterns of freight distribution caused by structural changes in logistics, such as port regionalization and the associated hinterland services development [20,21]. Mostly located within urban surroundings, inland PMBs need to deal with many different stakeholder groups pursuing different economic, social, and environmental interests in the port [21]. Unfortunately, not all of those stakeholder groups are equally aware of the positive contributions of an inland port, such as regional employment creation, value-added generation and more sustainable freight transport. Hence, embedding sustainability practices into the strategic plans and reporting about it should support societal stakeholders in perceiving the positive effects of the presence of an inland port in the urban region [22].

### 2.2. TBL Principle and Boundary Setting

At present, organizations face large pressures from different stakeholders to monitor and disclose information beyond mere financial performance. As a result, organizations are forced to rethink their idea of performance measurement in relation to their stakeholders, highlighting the need to include the social and environmental dimensions. However, including those extra dimensions into the business strategy and reporting practices requires new frameworks and standards to work with. The TBL concept is one of the most commonly used frameworks to assess economic, social, and environmental performance, and is considered as one of the best markers for defining the level of sustainability of an organization [23,24]. According to Gray and Milne [24], a strong TBL-based report covers and

elaborates on all dimensions equally, and provides linkages and trade-offs between them. However, more than 50% of the topic-specific GRI Standards belong to the social dimension [25], suggesting a rather unbalanced implementation.

Furthermore, a topic that has not yet received a lot of attention in literature is that of boundary setting in sustainability reports. Sustainability is foremost regarded as a global concept, which implies a certain degree of complexity when applied at the organizational level [24]. More specifically, sustainability issues related to social or environmental performance transcend the boundaries of the organization [26]. Unlike financial reporting boundaries, which rely on the principle of financial control, boundaries concerning sustainability performance explicitly need to consider impacts beyond full organizational control [15]. Setting the boundaries between the organization and its context is a multisided question, as it is not always a case of willingness but also of complexity. Broadening the reporting boundary to include, for example, the full port supply chain calls for data that are often not available in general or not available for the PMB specifically. Vice versa, for some indicators, broadening the scope does not necessarily mean gaining extra insightful information, for example, measuring gender equality on the level of the actors within the whole supply chain is less relevant. This research investigates how inland port stakeholders perceive (1) the idea of an equal contribution of the three TBL dimensions in a sustainability report, and (2) how boundaries should be defined for several categories of indicators.

### 2.3. Materiality and Stakeholder Inclusion

In the context of traditional financial reporting, the concept of materiality is shaped by both quantitative and qualitative aspects, all of them defined and written down in standards set by international organizations, imposed by governments, and used as a basic element of market-based investment decision-making [27,28]. For nonfinancial information, the narrative is very different, as this information cannot be directly and clearly valued in a market setting [29]. Materiality in the context of nonfinancial information therefore focuses on the external accounting of economic, environmental, social, and governance impacts towards stakeholders and not just investors as the principal 'market' (or 'audience') interested [27,30]. Stakeholders, as broadly defined by Clarkson [31], are "persons or groups that have or claim ownership, rights, or interests in a corporation and its activities, past, present, or future".

In practical terms, in order to provide a sustainability report in which all stakeholders are provided with relevant and comprehensive information, it is crucial to include those elements that reflect the interests and requests of each stakeholder group [27]. Reporting frameworks, such as GRI Standards, provide guidelines in order to handle the challenge of information asymmetry between organizations and their stakeholders, and to reduce the risk of not covering all material aspects. Therefore, a materiality analysis should be based on a participatory process, proactively engaging all stakeholders in an interactive dialogue to determine those aspects of information that are the most material [28]. This dialogue can entail different degrees of inclusion, reflecting gradual paths of stakeholder engagement. In our research, we have used and adapted the model of Friedman and Miles [32] to define the desired level of stakeholder inclusion in the process of sustainability reporting as perceived by stakeholders (Table 1). The seven different levels of involvement go from one extreme 'no or limited inclusion' to the other 'full inclusion'. It is often considered that the higher the level of inclusion, the greater the societal acceptance of strategic choices, and thus, the less unanticipated issues that can occur as committees have been ex ante included in the processes [33].

**Table 1.** Seven different levels of stakeholder inclusion.

| No or limited inclusion | | | | | Full inclusion | |
|---|---|---|---|---|---|---|
| **1.** | **2** | **3** | **4** | **5** | **6** | **7** |
| -Knowledge about the decisions. -One-way dialogue (e.g., briefing sessions, leaflets, corporate reports. | -Educating, explaining, and informing stakeholders. -One-way and/or two-way dialogue (e.g., verified corporate social reports, workshops). | -Stakeholders may advise. -Being heard before a decision. -Two-way dialogue (e.g., surveys, focus groups, interviews, etc.). | -Stakeholders provide conditional support. -Having an influence on decisions. -Multiway dialogue (e.g., bargaining, constructive dialogue). | -Collaboration/ partnership. -Some or joint decision-making power. -Multiway dialogue (e.g., strategic alliances, joint ventures). | -Minority representation of stakeholders in the decision-making process. -Multiway dialogue (e.g., board representation). | -Majority representation of stakeholders in the decision-making process. -Multiway dialogue (e.g., community projects). |

Table adapted by the authors from Friedman and Miles [32].

In the specific context of PMBs, especially those managing inland ports, the different types of ownership structures and the variety of involved stakeholder groups complicate the approach towards proper stakeholder engagement and the associated materiality analysis. For example, in most cases, the government does not only operate from a regulating role but also as a full or partial shareholder of the PMB, which creates a multitude of divergent objectives that need to be reached simultaneously. In parallel, all other stakeholder requests, such as those of port users, local communities, etc., need to be considered as well, as these parties heavily influence long-term port development plans [34–36]. The solution to this challenge does not limit itself to mere information dissemination strategies, but requires a high level of inclusion and a strong collaborative stakeholder approach in order to continuously increase the added value of the port and maintain its license to operate [36]. However, even though acceptance of strategic choices is mostly linked to a high level of stakeholder inclusion, it is important to investigate if the highest level of inclusion is desired by all stakeholders, as this also requires some investment in resources from their side. As a result, next to materiality, our research gathers complementary insights on the preferred degree of inclusion of the different stakeholder groups in the context of the development of a sustainability report.

In addition, by including stakeholders at the very beginning of the process and simultaneously taking account of the preferred level of inclusion, PMBs avoid that their potential efforts for developing a sustainability report are considered as a reactive solution with "greenwashing" reasons. As many papers highlight [37–39], the practice of developing a sustainability report should have the intention to inform and educate stakeholders, to create an environment of multiway dialogue and solution-thinking about pressing problems, to be able to measure and monitor past actions and future plans, etc. In other words, it is utterly important that the initiatives taken in light of improving the sustainability performance are seen as building blocks of the overall strategy of the organization.

## 3. Methodology

### 3.1. Case Study Selection: The Inland Port of Brussels

The definition of an inland port is polymorphic, meaning that it differs depending on certain features, such as its location, activities, connectivity, and role [12]. However, following the reasoning of Rodrigue and Notteboom [12], the Port of Brussels can be described as a comprehensive inland port as it encompasses inland waterway systems and handles a variety of traffic structures (inter alia construction materials, oil products, and containers). The Port of Brussels is a key node within the Trans-European Transport Network (TEN-T) and is located approximately 50 km from Antwerp, Europe's second largest seaport. Around 5 million metric tons are transshipped each year at the Port of Brussels. It makes part of the Seine-Scheldt basin connecting French, Belgian, Dutch, and German ports and waterway networks. As the port areas are located in the middle of the Brussels Capital Region (a so-called port "intra muros"), it is an interesting example of a metropolitan supporting

type inland port. The metropolitan supporting type shows a dominant urban and regional logistics functionality (e.g., construction industry logistics and distribution of consumer goods). Furthermore, port activities take place within or nearby residential areas, and are subject to continuous, outright contestation, hence reflecting the need for intensive and high-quality stakeholder interaction [40].

*3.2. Survey Design: Stakeholders and Topics*

For the purpose of this research, a survey was designed and disseminated to the major stakeholder groups of the Port of Brussels. The identification of the critical stakeholder groups was conducted making use of the general stakeholder theory [31,33] and complemented by the port-specific insights of Notteboom and Winkelmans [41], who identified four generic categories of stakeholders in the broader port environment. Initially, the following stakeholder groups were considered: personnel of the PMB, clients of the port (in this case tenants), broader society, and government agencies. The personnel of the PMB were considered as a salient stakeholder as they are entrusted with the daily managerial and operational tasks having a direct impact on the ongoing concerns of the port area. We decided to restrict the 'clients' category to tenants of the port as there is no direct nor intense commercial relationship of the PMB with inland shipping providers, as the tenants directly contract them and/or own or lease their own vessels for waterway transport. Furthermore, the broader society does not have a direct impact on the daily activities of the port, but at the same time experiences the positive/negative effects of the port activities on their own objectives, and actively influences political decision-makers. Finally, the 'government' stakeholder group is particular as these stakeholders are at the same time the principal shareholders/owners of the port but also need to act in accordance with the interests of the larger Brussels urban community, sometimes even being part of it. This breakdown into stakeholder categories was validated by several representatives of the Port of Brussels.

The survey started with a number of profile questions, followed by two general questions that probed for the uniform understanding and interpretation of the concepts of sustainability and sustainability reporting. The further content of the questionnaire was built upon comprehensive desk research and exploratory interviews with experts in the port environment, focusing on four large topics: (1) TBL concept, (2) materiality issues, (3) boundary setting, (4) stakeholder inclusion. Questions were expressed as close-ended affirmations, as this line of questioning helps in attaining a higher response rate [42], and is easier to code and analyze, while also avoiding mis-responses and misunderstandings with regard to the scales used [43,44]. To encode the questionnaire into an online version, we made use of the software program Qualtrics. Two pilot test runs took place by three representatives of the Port of Brussels, with modifications added after each pilot test before the final validation.

*3.3. Data Collection*

For the selection of the relevant respondents we made use of, and were dependent on, the stakeholder database of the Port of Brussels. Due to the EU data protection law (GDPR), the identification of relevant stakeholders, as well as the distribution of the survey accompanied by a cover letter, were handled by the Port of Brussels itself. The survey was sent out at the beginning of March 2019 and data were collected in the subsequent months, March and April 2019.

*3.4. Data Processing and Analysis*

SPSS was used to conduct statistical tests on the full amount of data. Furthermore, in order to analyze the question-specific data related to the TBL concept (see infra, Section 4.2), we based ourselves on the same method as applied in Calabrese et al. [7], namely the analytic hierarchy process (AHP) method. We opted for AHP because the questions probed for, in a way, subjective, qualitative information. The approach was deliberately chosen as it is a structured technique used to analyze complex multiple-criteria problems involving qualitative judgements. It splits the decision-making process into separate parts, each part assessing the importance of objects shown as a

paired comparison [45,46]. Every partial analysis consists of detailed in-depth results per stakeholder group, as well as broader overall conclusions.

## 4. Results

First, the most fundamental sample statistics are described before discussing the results of the analysis, which are divided into the different research areas of the case study.

### 4.1. Sample Statistics

Four stakeholder groups were initially defined (clients, personnel, broader society, and government). Unfortunately, no answers were received from the 'government' stakeholder group, which led to the removal of the entire group from the analysis. After personal contacts with several governmental representatives, who answered the survey, it became clear that they filled out the survey from their role as moderator of broader societal objectives. Most of them are also part of the local community as they reside in the Brussels Capital Region (i.e., the 'broader society' stakeholder group). In total, 105 responses were received, of which 31 needed to be deleted because of respondents that dropped out at the beginning of the survey. The division is as follows: clients (12), personnel (26), and broader society (36).

Even though the response rate of the 'clients' stakeholder group is not considerably high, the received responses are provided by the most relevant members of this group. The important objective was to reach those clients with the largest share in the total usage of the waterways of the Port of Brussels (a group of around 30 members) as they possess the largest control over the port. Furthermore, they also strengthen and confirm the strict terminology of an inland port as applied in this paper. In 75% of the received questionnaires, the client indeed makes frequent use of the waterways. Furthermore, the survey was sent out to all personnel members in order to not discriminate among the staff. However, being confronted on a regular basis with, and having an opinion/idea about, the concept of sustainability was a precondition to fill out the survey. In order to minimize the bias of self-selection, questions probing for the function of the respondent and his/her years of interest in sustainability were posed. Out of the 26 answers, only 2 come from personnel working for the department 'environment, health, and safety services'. All other received questionnaires come from respondents of various departments, showing an accurate representation of the organization. Also in terms of years of interest, a proper sample of respondents with a variety of experience and knowledge about the subject can be noted. The 'broader society' stakeholder group shows a good representation of several different stakeholder profiles.

According to Gay and Diehl [47], the type of research involved (descriptive, correlational, or experimental), defines the number of respondents needed to reach an acceptable response rate for analysis. They state that in the case of descriptive research, which is applied in this paper, a sample size covering 10% of the population is sufficient and 20% when the population is small. Given the number of received responses for each stakeholder group, in particular the clients, we can state that the sample size reflects a plausible proportion of the targeted population. Unfortunately, some questions were not fully completed by the respondents. For this reason, 'n' reflecting the absolute amount of answers will always be shown with each question being discussed. However, this has never led to the response rate being compromised. In all cases, the condition mentioned above remained fulfilled.

### 4.2. TBL Balance

The first part of the survey intended to investigate the perception of the stakeholders towards the content of a sustainability report. In other words, should each TBL dimension be equally elaborated on in a sustainability report? As mentioned in the methodology (Section 3.4), the AHP method was applied to analyze this data. Results are shown in Table 2. All stakeholder groups do not equally value the three dimensions of the TBL. In general, the findings suggest that the dimension 'environment' should encompass almost 50% of a sustainability report compared to the weak results of the social

dimension. Looking at the clients, the difference becomes even larger. For them the focus lay on the economic and environmental, less on the social dimension, in comparison with the other two stakeholder groups.

**Table 2.** Proportion of TBL concepts in a sustainability report.

|  | Economic | Social | Environmental |
|---|---|---|---|
| Personnel (*n* = 26) | 23% | 34% | 43% |
| Clients of the port (*n* = 12) | 31% | 20% | 49% |
| Broader society (including government) (*n* = 36) | 21% | 33% | 46% |

In light of current environmental discussion, the importance of the environmental dimension is not surprising. During the past decades, organizations have indeed put a larger focus on environmental issues, leading to a gap in the level of accuracy when comparing the conceptual and practical development of social and environmental performance [23]. Furthermore, looking at the clients' results we can further notice a dominance of the economic over the social dimension. A plausible explanation could be that financial reporting does not entirely satisfy the needs and demands of its relevant stakeholders, which are not shareholders, but are interested in and influenced by the actions of the organization. As mentioned by Deegan and Rankin [48], the users of an annual report are not exclusively limited to shareholders, but represent a varied cross-section of society. This supports the growing need for either extending the existing content of an annual report, focusing on more than just the financial side of an organization, or to create a separate sustainability report.

*4.3. Materiality Issues*

In order to explore those issues that are perceived as material by the different stakeholder groups, we made use of a Likert scale to grade each issue according to its level of importance, which is reflected by an indicator (ranging from 1 'not important' to 7 'very important'). The shortlist of possible material indicators was compiled based on literature research, the materiality analysis of the Port of Antwerp [49], and on own exploratory research in the context of the support for an industry working group (IAPH-PIANC Working Group 174), as well as recently completed applied research projects such as the PORTOPIA project [50]. In total, 38 indicators, adapted to the context of an inland port and divided into seven large domains, could be identified: economic (4), social (7), environmental (12), logistic and operational performance (5), mobility (3), port–city relationship (4), and satisfaction/perception (3). These seven domains were identified to cover the specific context of the Port of Brussels as not all indicators could immediately be categorized under the existing TBL dimensions.

The approach for the analysis was similar to the one applied in Font et al. [30], i.e., indicators were considered as Likert scale items on an interval level, to which parametric tests were applied. As stated in Font et al. [30] by Norman [51], regardless of the original data, for sample sizes greater than five, the central limit theorem underpins the condition that the means of those samples are generally normally distributed. For this analysis we compared the mean of each individual issue/indicator with a materiality baseline, which we set at a score of 5 (out of 7, i.e., 4 as neutral score), thus implying that the indicator is of importance. An extra t-test was applied to those indicators with a mean slightly lower than 5 to investigate if the difference was significant. The overview of means and related results of the list of indicators can be found in Table 3. Results show that for the stakeholder group 'personnel', 37 out of 38 indicators comply with the materiality baseline, meaning that they can be considered as material issues. Only the indicator 'staff turnover' is regarded not important enough to include into a sustainability report. The clients show the same results as the personnel and add the indicator 'gender equality' as unimportant issue. In conjunction with the other results, the broader society group also perceives two indicators as not worthy to pay attention to: 'staff turnover' and 'level of difficulty hiring staff'. Remarkably, all nonsignificant indicators are part of the social dimension.

**Table 3.** List of indicators.

| | Personnel | Clients | Broader Society | Average |
|---|---|---|---|---|
| **Economic** | | | | |
| Investment volume | 5.2 | 4.6 | 5.0 | 4.9 |
| Procurement practices | 5.6 | 4.4 | 5.4 | 5.1 |
| Indirect economic impact (added value) | 5.0 | 5.2 | 5.3 | 5.2 |
| Direct economic impact (added value) | 5.8 | 5.3 | 5.4 | 5.5 |
| **Social** | | | | |
| Staff turnover | **4.3** | **4.1** | **4.3** | 4.2 |
| Level of difficulty hiring staff | 4.5 | 4.4 | **4.3** | 4.4 |
| Gender equality | 4.9 | **3.3** | 5.2 | 4.5 |
| Indirect employment | 5.5 | 4.8 | 5.4 | 5.2 |
| Direct employment | 5.7 | 4.9 | 5.6 | 5.4 |
| Health and safety at work | 5.4 | 5.5 | 5.2 | 5.4 |
| Education and training | 5.4 | 5.4 | 5.4 | 5.4 |
| **Environmental** | | | | |
| Odors | 5.3 | 5 | 5 | 5.1 |
| Dredging | 5.9 | 4.8 | 5.3 | 5.3 |
| Port expansion | 5.5 | 4.9 | 5.8 | 5.4 |
| Biodiversity/nature | 6 | 5.1 | 5.4 | 5.5 |
| Noise | 5.9 | 5.3 | 5.5 | 5.6 |
| Ship waste | 6.2 | 5.8 | 6.1 | 6.0 |
| Water consumption | 6.5 | 5.6 | 5.9 | 6.0 |
| Green policy and actions | 6.5 | 5.6 | 6 | 6.0 |
| Port waste | 6.3 | 5.8 | 6.1 | 6.1 |
| Ship discharges to water | 6 | 6.2 | 6.1 | 6.1 |
| Energy consumption | 6.7 | 5.8 | 6.1 | 6.2 |
| Quality/emissions (air, water, soil) | 6.7 | 6.1 | 6.1 | 6.3 |
| **Logistic and Operational Performance** | | | | |
| Spatial productivity per quay meter | 5.5 | 5 | 5.3 | 5.3 |
| Area usage of the different sectors | 5.5 | 5.1 | 5.2 | 5.3 |
| Throughput per quay meter | 5.6 | 5.2 | 5.5 | 5.4 |
| Seaport connectivity | 5.6 | 5.4 | 5.9 | 5.6 |
| Intermodal connectivity | 5.9 | 5.7 | 6.2 | 5.9 |
| **Mobility** | | | | |
| Modal split commuter traffic | 5.7 | 5.1 | 5.1 | 5.3 |
| Road congestion around the port area | 5.8 | 5.6 | 6 | 5.8 |
| Future actions | 6.1 | 5.8 | 6.1 | 6.0 |
| **Port-City** | | | | |
| Integration of the port into the Trans-European waterways' framework | 6 | 4.8 | 5.7 | 5.5 |
| Integration of the territorial management of the Canal area in present and future plans of the port | 5.8 | 5.6 | 6.1 | 5.8 |
| Integration of the port into plans of new developments on Federal and Regional level (Flanders/Wallonia) | 6.2 | 5.6 | 5.7 | 5.8 |
| Integration of present and future port activities into the metropolitan environment | 6.3 | 5.5 | 6 | 5.9 |
| **Satisfaction/Perception** | | | | |
| User/client satisfaction | 5.7 | 4.9 | 5.4 | 5.3 |
| Employee satisfaction | 5.8 | 4.8 | 5.3 | 5.3 |
| Local communities' perception | 5.8 | 5.1 | 5.4 | 5.4 |
| *N =* | **22** | **12** | **30** | **64** |

Underlined numbers = indicators with a mean lower than 5; Numbers in red = indicators with a mean significantly different from 5 ($p < 0.05$).

In order to have a more general overview and to shed light on some of the higher-level tendencies, additional descriptive statistics were applied to the data. More specifically, the technique of quartiles was used to have a more in-depth look at the distribution of the responses of all stakeholder groups per indicator. A tendency to assign higher importance to the economic indicators can be observed. However, clients seem to be more interested in general high-level economic indicators, like (in)direct economic impact, than those linked to the daily businesses of the port. Overall, the social indicators are valued less important. The distribution of answers from personnel for the indicators 'staff turnover' and 'level of difficulty hiring staff' is very spread out, showing few persons for whom this issue is not even important enough to be mentioned in a sustainability report. The same phenomenon holds true for the indicators 'health and safety at work' and 'education and training'. Among the answers of the personnel there are some unexpected tendencies, with people showing no interest in the inclusion of these indicators in a sustainability report. Taking a closer look, it seems to be employees with the most recent interest in the topic 'sustainability' (compared to the other respondents) that gave the lower scores. Nevertheless, these indicators are given high overall scores by the clients and broader society. Additionally, more than 50% of the clients are not interested in gender equality in the organization, in contrast with the results of the broader society, where 70% of the answers are equal or above the score of 5.

Furthermore, as expected, the environmental indicators are given a high level of importance by all stakeholders. Issues such as energy consumption and quality/emissions almost even reach the maximum score of 7. The same level of importance is also acknowledged by academics with papers looking into these specific research areas [52,53]. Results of the 'broader society' stakeholder group make us believe that the survey has been answered by many stakeholders with a professional 'urban development' background. The indicators 'spatial productivity per meter of quay' and 'seaport connectivity', as well as issues related to the port–city relationship and the integration of the port into regional, national, and international transport and infrastructure development plans are given very high scores. In contrast, the answers of the clients regarding the integration of the port into the Trans-European waterways' framework are very divided, with one of the lowest means of the exercise (see Table 3). A possible explanation is that most waterway users in Brussels use fixed, specific connections that already function very well and are not hampered by infrastructural bottlenecks on the wider EU waterway network. Finally, developing an indicator that reflects the perception and satisfaction of the different stakeholders is regarded as very important by the personnel of the PMB. It shows awareness of the increasing influence that external stakeholders have on port activities and development, making it a necessity to invest in superior stakeholder management.

*4.4. Boundary Setting*

Besides defining the materiality of several indicators, setting the boundaries is another complex dilemma that needs to be determined when preparing a sustainability report. The respondents were asked to select for each dimension (economic–social–environmental) the boundary that they deemed of high relevance for the PMB to report on (based on what is feasible in the PMB's present situation). The boundaries were defined by the authors based on the organizational and geographical/operational features of the port, following the approach and philosophy in Archel, Fernandez, and Larrinaga [54]. If the PMB should report an indicator on both the level of the PMB itself as well as on the level of the port cluster separately, the respondent needed to tick both boundaries A and B. If the respondent was of the opinion that the PMB should measure an indicator only on the level of the port cluster (thus not for the PMB separately), only boundary B should be ticked. Table 4 shows the results of this exercise.

Although there is accordance on the contribution of environmental indicators in a sustainability report, a small difference between the stakeholder groups can be noticed when it comes to defining the relevant boundaries. For the clients and community stakeholders, it seems to be very important that environmental indicators are calculated comprising all activities of the supply chain, in contrast with the view of the personnel who considers boundaries A and/or B as sufficient. A similar reasoning

applies in large part to the social dimension, with clients and broader society stakeholders focusing on boundary C. In general, we can state that the personnel prefers to first put the focus on the own organization with attempts to broaden the scope, in comparison to the two other stakeholder groups, for whom the boundary of the organization is a minimum condition and who prefer to see indicators calculated on broader boundaries. For all stakeholder groups, the economic dimension should be analyzed with a focus on the first two boundaries, thus focusing on those actors that can have a direct impact on the going concern of the organization.

**Table 4.** Boundaries per domain perceived by the different stakeholder groups.

| | **Economic** | | | | |
|---|---|---|---|---|---|
| | A | B | C | D | No opinion |
| **Personnel (n = 22)** | 55% | 59% | 32% | 23% | 5% |
| **Clients/users (n = 12)** | 42% | 58% | 17% | 8% | 8% |
| **Broader Society (n = 30)** | 43% | 53% | 30% | 23% | 20% |
| | **Social** | | | | |
| | A | B | C | D | No opinion |
| **Personnel (n = 22)** | 55% | 41% | 45% | 23% | 5% |
| **Clients/users (n = 12)** | 33% | 25% | 58% | 17% | 8% |
| **Broader Society (n = 30)** | 37% | 43% | 57% | 10% | 20% |
| | **Environmental** | | | | |
| | A | B | C | D | No opinion |
| **Personnel (n = 22)** | 64% | 55% | 36% | 27% | 5% |
| **Clients/users (n = 12)** | 42% | 42% | 33% | 50% | 8% |
| **Broader Society (n = 30)** | 40% | 50% | 43% | 43% | 20% |

A = Port Managing Body organization; B = port area/cluster (including industry/logistics and including the hinterland interface); C = local/regional community; D = impact of upstream and downstream supply chain activities taking place outside the port borders and beyond the local/regional community.

### 4.5. Stakeholder's Willingness for Inclusion

The creation of mutual responsibility between the organization and its stakeholders is necessary to maintain the license to operate as expectations can be managed and aligned with each other [55]. However, such inclusion also requires time and efforts from the stakeholders. Eventually, stakeholders would prefer a level of inclusion for which the cost–benefit balance will be neutral or positive.

Based on the adapted version of the model of Friedman and Miles [32] (see Section 2.3, Table 1), each respondent was asked to indicate the desired level of inclusion during the process of sustainability reporting. Each level is linked to a number, ranging from 1 (no inclusion) to 7 (full inclusion). Table 5 presents the mean and related standard deviation. All stakeholder groups have a mean of almost 5, which corresponds with a level of collaboration, joint decision-making, and multiway dialogue. However, looking at the standard deviations, this result should be interpreted with caution. A more in-depth analysis of the results shows that the majority of the clients gave a score of 5 or 6, but two outliers could be identified with a score of 1 and 2. In all probability, this can be explained by the type, size, portfolio of activities, etc. of the organizations in question. The distributions of the personnel and broader society are quite comparable, with answers equally divided between 3 and 7. This shows that the personnel of the port is aware of the importance of strong stakeholder inclusion, not only to anticipate demands and expectations, but also to create a learning environment stimulated through mutual interaction. Although being important for the urban region, the PMB is not a dominant player and does not have the ultimate bargaining position, which causes long lead times to realize projects (5 to 10 years). For this reason, the PMB should always keep its stakeholder strategy on point.

**Table 5.** Preference of stakeholder inclusion by the different stakeholder groups.

|  | Mean | SD |
|---|---|---|
| Personnel (*n* = 21) | 4.95 | 1.564 |
| Clients/users of the port (*n* = 11) | 4.64 | 1.690 |
| Broader society (*n* = 30) | 4.90 | 1.213 |

Furthermore, a question concerning preferred communication tools was also posed to the external stakeholder groups (clients and broader society). Figure 1 shows that a survey and a workshop are the desired tools by both stakeholder groups. In a second phase, more face-to-face meetings are favored, with a preference for a one-on-one conversation by the clients and a focus group by the community stakeholders. Being consulted every six months is the frequency chosen by the majority of both groups.

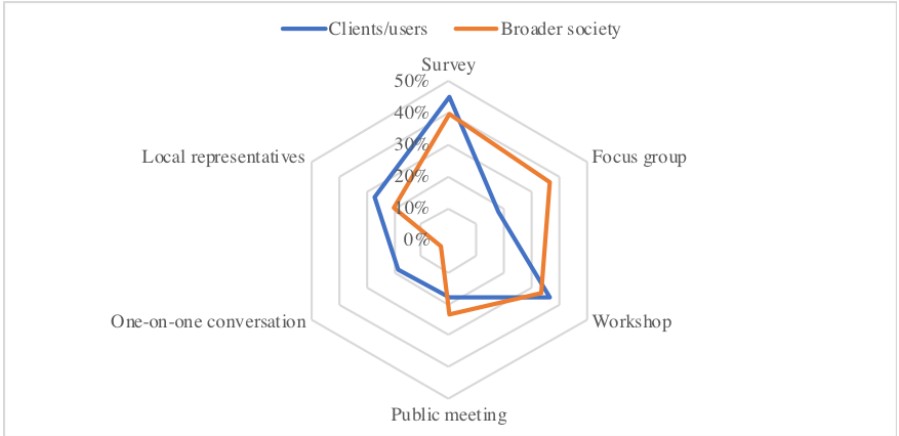

**Figure 1.** Communication means preferred by clients/users and broader society.

## 5. Discussion

Table 6 gives an overview of the different discussed topics by linking the literature and the survey results and adding concluding reflections. These reflections form the synthesis of the theoretical and practical perspective on some important topics at present with regard to sustainability reporting. Some additional explanation is provided in the sections below.

### 5.1. TBL Balance and Materiality Issues

More purpose-driven employees, risk education, monitoring long-term risks, etc. as internal, sometimes organization-specific advantages, cover just one part of the full set of advantages of sustainability reporting. Societal aspects, more externally driven, such as an increase in transparency, enhanced reputation, improved market position, improved stakeholder relations, etc. complete the list and characterize the growing importance of the practice [1,3,56]. Although sustainability reporting has already proven to be of value for organizations, developing such a report is not straightforward as social and environmental performances are very difficult to quantify and are unique to each organization. Issues that are considered material by stakeholders will differ between organizations, for example those situated in developing or developed countries, as local environmental and social requirements differ and as most likely another interpretation of the ideal ongoing concern strategy exists. These different conditions and playing fields for organizations will also be translated into different 'optimum' TBL balances per organization. However, it is important that each TBL dimension contains a minimum level of content determined by the highest common denominator in terms of objectives of the different stakeholder groups. Even though every situation is unique and needs proper judgement, the need for sector-specific guidelines is high as they could help in defining that necessary minimum level of compliance.

**Table 6.** Conclusive table.

| | Sustainability Reporting Literature | Survey Results (Port of Brussels) |
|---|---|---|
| **TBL balance** | ➢ Increased importance of reporting on more dimensions than the mere financial performance of an organization.<br>➢ A strong TBL report equally values all dimensions and identifies trade-offs and linkages between economic, social, and environmental performance.<br>➢ More than 50% of the GRI Standards belong to the social dimension. | ➢ The 3 dimensions of the TBL are not equally valued by the stakeholder groups.<br>➢ The environmental dimension should cover up to 50% of the content of a sustainability report, according to all stakeholders.<br>➢ Relatively weak results for the social dimension.<br>➢ Clients put more importance on the economic compared to the social dimension. |
| | **Reflection**<br>The optimal balance between the three dimensions of the TBL is context-dependent and should take account of all needs of the different stakeholder groups. | |
| **Materiality issues** | ➢ Audience shifted from being only shareholders to all stakeholders.<br>➢ More difficult to 'value' (quantify) information of the environmental and social dimension. | ➢ All nonsignificant indicators are part of the social dimension.<br>➢ All stakeholder groups stipulate environmental indicators as most material.<br>➢ No significant differences between materiality preferences of the stakeholder groups. |
| | **Reflection**<br>Need for sector-specific frameworks and guidelines on sustainability reporting.<br>For each dimension (eco, soc, and env) the highest common denominator of the demands of the different stakeholder groups in terms of total material topics should be investigated. | |
| **Boundary setting** | ➢ No fixed definition of boundary setting and its unit of measurement for a PMB.<br>➢ No fixed guidelines stipulating the required boundary per indicator.<br>➢ In general and at present, boundary setting is still mainly based on the consideration of financial control of the organization. | ➢ The PMB personnel prefers to put a first focus on the own organization, with attempts to broaden the scope compared to the clients and broader society stakeholders for whom the boundary of the organization is a minimum ambition when it comes to the environmental and social dimension of sustainability reporting.<br>➢ Overall alignment between stakeholder groups on the boundaries for economic indicators. |
| | **Reflection**<br>Boundary setting depends on the used unit of measurement, e.g., operational vs. organizational boundaries.<br>Broadening boundaries can lead to the inclusion of negative performance outside the own organization. | |
| **Stakeholder's willingness for inclusion** | ➢ Stakeholder inclusion should go beyond dissemination of information and should reach a strong level of collaboration: stronger stakeholder inclusion improves the outcome of the sustainability process. | ➢ All stakeholder groups prefer the same level of inclusion (no maximum) in the process of sustainability reporting. |
| | **Reflection**<br>Full inclusion is not preferred by the stakeholders themselves, as the costs would outweigh the perceived benefits. | |

### 5.2. Boundary Setting

Boundary setting in the context of financial reporting is known and well-defined, i.e., based on the concept of financial control. Sustainability reporting on the other hand involves more than the economic aspects of an organization, meaning that the principle of boundary setting should consider more than only those aspects under financial control. Based on the unit of measurement, several approaches to sustainability boundary setting can be articulated. Two commonly used units of measurement are based on organizational and operational features, respectively horizontal and vertical boundary setting. In the context of ports, operational boundaries are often applied. Furthermore, based on the results, we observe that the PMB is more willing to broaden an indicator's boundary when this results in potentially showcasing more positive news, cfr. social dimension. In contrast, when broadening the boundary is associated with the potential publication of negative news, the PMB will be reluctant, cfr. environmental dimension.

Boundary setting is an important element of sustainability reporting as, when well-defined, it can form the link between the micro organizational level and the macro level contribution of the port cluster managed by the PMB to sustainable development. These insights will allow identifying and monitoring high level integration opportunities [57], as well as possible negative effects caused by seaports but 'absorbed' by inland ports. Inland ports with a strong license to operate might provide tangible benefits to seaports, as the latter rely on inland ports to achieve more sustainable hinterland logistics, and thus improving their own sustainability impacts. This in contrast with the inland ports in question, as they are responsible for the last mile of the transport chain and thereby confronted with the less attractive transport mode: road. In light of the research presented in this paper we also identified the availability of resources as a major barrier for inland ports to engage in sustainability reporting; deeper collaboration between inland and seaports within the same network/supply chains seems warranted. The main challenge of such collaboration is most likely the definition of sustainability reporting outputs, which still appeal to local stakeholders (e.g., local communities).

### 5.3. Stakeholder's Willingness for Inclusion

Although the level of stakeholder inclusion is mostly positively correlated with societal acceptance of strategic choices made on the level of the organization [33], a specific element related to cognitive or information overload should be considered [58]. When more stakeholders get involved in organizational and decision-making processes, this also means a larger exposure to new information and a potential increase in complexity of institutional partnerships. However, every organization, and by extension institutional system, is limited by its size, inter alia defined by its human and financial resources, which corresponds with a certain level of cognitive saturation. More specifically, above a certain level the benefits of additional information will exceed the costs of processing it. When stakeholders within the system ignore this limitation, chances increase that they will be confronted with a cognitive overload at some point, leading to an overall loss of value in terms of the information presented. In general, it is an exercise of balancing with a saturation level defined by the stakeholders' and organizations' particular characteristics. Therefore, the level of cognitive saturation of the Port of Brussels's stakeholder system will probably be lower as compared to a large seaport, such as the Port of Antwerp, and thus matching another optimal number of stakeholders to be included.

## 6. Conclusions

Inland ports operate under different, sometimes more extreme, circumstances than seaports, specifically when it comes to resource availability (human and financial), and the multitude of stakeholders influenced by the presence of the port in an urban context. This research contributes to existing literature as it approaches the concept of sustainability reporting in the context of inland port managing bodies from the perspective of three of their important stakeholder groups (personnel, clients, and broader society). The research results of the paper are of interest to academics and practitioners,

as well as policy makers. The Port of Brussels has been used as a case study to discover potential gaps in expectations between different stakeholder groups with regard to sustainability (reporting) and its subdimensions: materiality, boundary setting, and stakeholder inclusion. The research shows how a materiality analysis and adequate boundary setting can play crucial roles in addressing the demands and needs of the different stakeholder groups, hence creating a better understanding and future progress in managing expectations. A sustainability report as a result of various exercises on all TBL dimensions can be regarded as the epitome of the 'sustainability DNA' of the port managing body, and will be of importance in strengthening its license to operate.

Limitations of the research concern the focus on one case study and the relatively small sample size. Due to the adoption of a holistic approach, based on the Port of Brussels as a case study, generalization should be made with caution. Insights of this paper provide only a first step in the development of a framework around sustainability reporting for inland port managing bodies and potentially also smaller seaports close to urban environments. Nevertheless, based on the findings of this exploratory research, several applications and possibilities for future research can be defined. First, this research can serve as a basis and information source for inland port managing bodies when analyzing the potential of sustainability disclosure, given the lack of examples in this domain. The analysis combines ideas found in literature with evidence in practice, which demonstrates the mutual benefits of collaborative research between academics and real-life practitioners. Second, as the focus of this research lies on the Port of Brussels as a metropolitan inland port, it would be interesting to also replicate the analysis to other types of inland ports, such as industry supporting types, or to other types of economies, such as emerging economies. For emerging economies in particular, the role of the port managing body when it comes to sustainability reporting (given different port governance frameworks), as well as the applied boundaries in terms of issues and stakeholders to be included, might differ substantially. Third, the survey instrument for this research has been developed in a way that it can easily be applied and adapted to the specific terms of other inland port cases or even smaller seaports next to or within urban environments, without losing the robustness of the survey. Additionally, the current insights could be complemented with in-depth interviews or focus groups with stakeholders to analyze more in-depth the underlying reasons behind some of the specific results of the survey.

**Author Contributions:** M.G.: conception and design of the study, methodology, acquisition of data, analysis and interpretation of the data, drafting the manuscript and data presentation, revising it critically for important intellectual content, final approval of the version to be published; M.D.: survey design and quality control, writing in parts related to discussions, quality control. All authors have read and agreed to the published version of the manuscript.

**Funding:** This research did not receive external funding.

**Conflicts of Interest:** The authors declare no conflict of interest.

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
