# Peer review of "Sustainability Reporting for Inland Port Managing Bodies: A Stakeholder-Based View on Materiality"

_sustainability, doi:10.3390/su12051726_

Round 1
Reviewer 1 Report
The authors correctly point out that often companies and scientific literature use corporate social responsibility (CSR) reporting and sustainability reporting as interchangeable terms. Although similar concepts, it is not a sound practice, since the CSR and sustainability of a business have slightly different meaning. To simply put it, whereas CSR focuses on activities outside of the core activities of a company to “pay debt” to this company’s external stakeholders such as society, which is affected by the business activities of the given company. However, sustainability in turn focuses on mitigating the ongoing negative impacts of this given company’s activities. It would be recommendable to explore this difference and contribute the results to the scientific literature. Bansal and Song (2017) have studied this difference on a general perspective, but they call for more studies that examine these concepts as similar but with different purpose. Their article is worth looking into when doing sustainability research.
Despite few minor typing errors (e.g., page 2 row 43, it should be KPMG not KMPG), the text is well written. The research is relevant, since it thrives to bridge the gap between theory and practice with a straight-forward approach. The results of the study are beneficial for the scientific community in questioning the balance of the three aspects in the Triple Bottom Line theory. If the mentioned short-comings are revised, the article should be accepted to additional review in order to publish it in the journal.
Weakness of the study are related to the amount of answers to completed questionnaire (overall it is satisfactory/good, but e.g. individually clients/users is quite low). Please motivate the low amount of responses, e.g. by how you selected respondents. Also it would benefit, if authors would reveal what kind of interaction project had with inland sea port, and if you had development project together, then it would increase the validity of study to describe, what kind of response people at sea port had on this study results. It would be worthwhile to include questionnaire form to the appendix of this manuscript, however, if it is following closely reported items, then it could be left not to be included. Did you have any free text response part in survey? Should you report these results?
Do check that in all lists within main text you are using "and" word between last and second last item. Currently, this is not the situation. Secondly, within text figures and tables should be written with starting capital letter, even in the middle of sentence (Table 1 vs. table 1).
It would indeed benefit, if authors would include Lam et al. (2019), Li et al. (2019) and Dey et al. (2011) in their literature review. This of course, if authors see them beneficial (e.g. inland ports are in principle similarly good for environment as are railway based dry ports).
References
Bansal, P., & Song, H. C. (2017). Similar but not the same: Differentiating corporate sustainability from corporate responsibility. Academy of Management Annals, 11(1), 105–149. https://doi.org/10.5465/annals.2015.0095
Dey, A., LaGuardia, P. and Srinivasan, M. (2011). Building sustainability in logistics operations: a research agenda. Management Research Review, 34:11, pp. 1237-1259.
Lam, J. S. L. & Li, K. X. (2019). Green port marketing for sustainable growth and development. Transport Policy, 84, pp. 73-81.
Li, W., Hilmola, O-P. & Panova, Y. (2019). Container sea ports and dry ports: Future CO2 emission reduction potential in China. Sustainability, 11:6, 1515; https://doi.org/10.3390/su11061515.
Reviewer 2 Report
This a well-written paper on the sustainability reporting for inland ports. Authors conduct a survey to understand stakeholder's perception on sustainability components, materiality, boundaries and inclusion in the sustainability reporting procedures and deliverables. There are strong findings from the empirical research. I am very positive about the submission. Yet, following comments should be well addressed to improve the paper. I am looking forward to revision;
) You can discuss the effects of sustainability reporting on marketing.
) I understand that there is no report for inland ports. But you should also comment on the available reports of hinterlands, dry ports in section 2.1, if available.
) More insights from seaport sustainability reports can be obtained. Are there similarities to inland ports? What are differences?
) I would like to bring authors' attention to following relevant study as energy efficiency is noted to be one of highest important indicators on average. I think you should incorporate following study in your paper;
A review of energy efficiency in ports: Operational strategies, technologies and energy management systems. Renewable and Sustainable Energy Reviews, 112, 170-182.
) I also have some methodological comments. AHP is noted to be used in TBL balance. But traditionally, AHP is used for selecting an alternative considering a criteria set. It requires further explanation how AHP is used in this study.
) For table 2, the reviewer cannot read the meaning of underlined numbers. It is probably t-test result, but it is not mentioned in the paper.
) There are following studies that integrate aspects of sustainability (environmental, economic and social) through collaboration between inland ports and other parties. These following studies could be incorporated in your paper;
The multi-port berth allocation problem with speed optimization and emission considerations. Transportation Research Part D: Transport and Environment, 54, 142-159.
Strategic integration of the inland port and shipping service for the ocean carrier. Transportation Research Part E: Logistics and Transportation Review, 110, 90-109.
) Minor language and misc. suggestions;
pg 1, l 41, "," not "."
Delete "for example, ..." on pg 4 l 152
pg 10, l 362, you mention environmental indicators, then you talk about road congestion which is not in the environmental indicator list.
pg 11, l 369, "an agreement"
Round 2
Reviewer 2 Report
My comments are well addressed. The paper can be accepted.